# Emergent geometry and duality in the carbon nucleus

Shihang Shen [1], Serdar Elhatisari [2,3], Timo A. Lähde [1,4], Dean Lee [5] ✉, Bing-Nan Lu[6] & Ulf-G. Meißner [1,2,4,7]

The carbon atom provides the backbone for the complex organic chemistry composing the building blocks of life. The physics of the carbon nucleus in its predominant isotope, $^{12}$C, is similarly full of multifaceted complexity. Here we provide a model-independent density map of the geometry of the nuclear states of $^{12}$C using the ab initio framework of nuclear lattice effective field theory. We find that the well-known but enigmatic Hoyle state is composed of a "bent-arm" or obtuse triangular arrangement of alpha clusters. We identify all of the low-lying nuclear states of $^{12}$C as having an intrinsic shape composed of three alpha clusters forming either an equilateral triangle or an obtuse triangle. The states with the equilateral triangle formation also have a dual description in terms of particle-hole excitations in the mean-field picture.

The physics of the $^{12}$C nucleus is a fascinating subject with a long and fabled history[1,2], and recent groundbreaking experimental results have provided hints of new states with exotic structures[3–13]. However, the underlying structures of several nuclear states of $^{12}$C remain without a consensus of agreement, and answers to such questions would provide deep insights into the emergent correlations relevant to nuclear binding and the panoply of possible structures that may appear in other nuclear systems. The most famous example is the case of the so-called Hoyle state, and its hypothetical rotational band partners. The Hoyle state is a narrow resonance, whose close proximity to the energy threshold for three alpha particles greatly enhances the reaction rate of the triple-alpha process, which is key to the production of carbon in evolved, helium-burning stars[14,15]. Much progress has been made in understanding the spectrum of $^{12}$C including the Hoyle state, in theoretical studies using the no-core shell model[16,17], symmetry-adapted no-core shell model[18], shell model[19], Monte Carlo shell model[20], quantum Monte Carlo simulations (QMC)[21], replica exchange MC (RXMC)[22], antisymmetrized molecular dynamics (AMD)[23–25], fermion molecular dynamics (FMD)[26], density functional theory[27–29], Bose-Einstein condensate (BEC) wave functions[30–32], alpha cluster models (ACM)[26], and nuclear lattice effective field theory (NLEFT)[33,34].

There are two main impediments to reaching definitive conclusions about the structure of the low-lying $^{12}$C states. The first is the inability to perform calculations that can handle strong multi-particle correlations. The second is the inability to measure the detailed spatial correlations required to determine the intrinsic structure of the twelve-particle wave function. In this work we address both problems. We perform unconstrained lattice Monte Carlo simulations using the framework of NLEFT[35,36], including all possible multi-particle quantum correlations. The lattice results are in good agreement with experimental data. Furthermore, we determine the full twelve-particle correlations and use a model-independent density projection to determine the intrinsic structure of each nuclear state.

## Results and discussion
### Spectrum and electromagnetic properties
We present results obtained from nuclear lattice simulations using two different interactions. The first is an interaction that is independent of spin and isospin. We call it an SU(4) interaction due to the symmetry with respect to the four nucleonic degrees of freedom. The second interaction is based on the framework of chiral effective field theory, carried out to next-to-next-to-leading order (N2LO). We have

[1]Institut für Kernphysik, Institute for Advanced Simulation, Jülich Center for Hadron Physics, Forschungszentrum Jülich, D-52425 Jülich, Germany. [2]Helmholtz-Institut für Strahlen- und Kernphysik and Bethe Center for Theoretical Physics, Universität Bonn, D-53115 Bonn, Germany. [3]Faculty of Natural Sciences and Engineering, Gaziantep Islam Science and Technology University, Gaziantep 27010, Turkey. [4]Center for Advanced Simulation and Analytics (CASA), Forschungszentrum Jülich, D-52425 Jülich, Germany. [5]Facility for Rare Isotope Beams and Department of Physics and Astronomy, Michigan State University, East Lansing, MI 48824, USA. [6]Graduate School of China Academy of Engineering Physics, Beijing 100193, China. [7]Tbilisi State University, 0186 Tbilisi, Georgia. ✉ e-mail: leed@frib.msu.edu

calculated the $^{12}$C spectrum up to excitation energies of about 15 MeV. The results are plotted in the left panel of Fig. 1 for different values of the angular momentum and parity. For comparison, we show the experimental data (black stars)[37]. The error bars are one standard deviation estimates due to stochastic errors and Euclidean time extrapolation uncertainties. The lattice results using the SU(4) and N2LO chiral interactions are in good agreement with each other and in good agreement with experimental data. The triangular shapes surrounding the data points indicate intrinsic shapes and are explained later in our discussion. In the right panel of Fig. 1, we show the form factors for the ground state and the Hoyle state in the top figure (b), and the transition form factor from the ground state to the Hoyle state in the bottom figure (c). For comparison, we show the experimental data for the ground state and transition form factors[5,38–40], and the agreement is fairly good.

In Table 1, the energies and radii of $0_1^+, 0_2^+, 0_3^+, 2_1^+$, and $2_2^+$ states are shown in comparison with other theoretical calculations[22,26,31,32] and experimental data[37,41]. Our $J_n^\pi$ notation indicates the angular momentum $J$, parity $\pi$, and ordinal number $n$. The NLEFT energies and radii agree very well with empirical results. Since the $0_2^+, 0_3^+$ and $2_2^+$ states are unbound or nearly unbound with respect to the three-alpha threshold, the radii are calculated with the corresponding state is placed in a periodic cube with length $L = 14.8$ fm. The first error bars are one standard deviation estimates due to stochastic errors and Euclidean time extrapolation uncertainties. The second error bars are an estimate of systematic errors due to broken rotational invariance from the finite periodic volume and nonzero lattice spacing. Additional systematic errors due to the choice of interaction are described in the Supplementary Information. In Table 2, the electric quadrupole moments of the $2^+$ states and electric transition rates involving the $0_1^+, 0_2^+, 0_3^+$, and $2_1^+$ states of $^{12}$C obtained by NLEFT are given in comparison with other theoretical calculations[26,42] and experimental data[43,44]. While the quadrupole transitions have errors due to broken rotational invariance, the overall agreement with empirical results is good.

## Intrinsic structures

We compute the intrinsic structure of the nuclear states in the following manner. We first select pinhole configurations which contain exactly three spin-up protons. For each spin-up proton, we locate the closest spin-down proton, spin-up neutron, and spin-down neutron. We identify these four nucleons as an alpha cluster, and determine its center of mass. In this manner, we locate the positions of three alpha clusters for each nuclear state of $^{12}$C. The root-mean-square (RMS) matter radii of the alpha clusters defined in this manner range from 1.65 fm to 1.71 fm for the low-lying states in $^{12}$C. This is very close to the matter radius of an isolated alpha particle using the same interactions, $r_\alpha = 1.63$ fm, and this gives us confidence that we are indeed identifying alpha clusters. While the process of identifying alpha clusters is not unique, the method we are using here is defined unambiguously from the twelve-nucleon spatial probability distribution. So long as we are not probing very short-distance correlations between nucleons, the twelve-nucleon spatial probability distribution is model independent. Therefore, our definition of the alpha cluster geometry is also model independent.

The three alpha clusters that we have identified define a triangle in three-dimensional space with interior angles $\theta_1$, $\theta_2$, and $180° - \theta_1 - \theta_2$. In the left panel of Fig. 2, we show the probability distributions as a function of $\theta_1$ and $\theta_2$ for (a) the $0_1^+$ ground state, (b) $0_2^+$ Hoyle state, (c)

### Table 1 | Energies and charge radii of $^{12}$C

|  | NLEFT | FMD | α cluster | BEC | RXMC | Exp. |
|---|---|---|---|---|---|---|
| $E(0_1^+)$ | −91.6 (1) | −92.6 | −89.6 | −89.5 | −88.0 | −92.2 |
| $E(0_2^+)$ | −85.7 (1) | −83.1 | −81.7 | −81.8 | −81.4 | −84.5 |
| $E(0_3^+)$ | −82.7 (1) | −80.7 | −79.2 | – | −79.0 | −81.9 (3) |
| $E(2_1^+)$ | −86.9 (1) (1.5) | −87.3 | −87.0 | −86.7 | – | −87.7 |
| $E(2_2^+)$ | −82.5 (1) (2.1) | −80.8 | −80.4 | −80.5 | – | −82.3 |
| $r_c(0_1^+)$ | 2.54 (1) | 2.53 | 2.54 | 2.53 | 2.65 | 2.47 (2) |
| $r(0_2^+)$ | 3.45 (2) | 3.38 | 3.71 | 3.83 | 4.00 | – |
| $r(0_3^+)$ | 3.47 (1) | 4.62 | 4.75 | – | 4.80 | – |
| $r(2_1^+)$ | 2.42 (1) (1) | 2.50 | 2.37 | 2.38 | – | – |
| $r(2_2^+)$ | 3.30 (1) (4) | 4.43 | 4.43 | – | – | – |

The energies and charge radii $r_c$ (or point radii $r$) of $^{12}$C are calculated with NLEFT using the SU(4) interaction and compared to calculations from FMD[26], ACM[26], BEC[31,32], and RXMC[22] as well as experiment[37,41]. All energies are in MeV and radii in fm. For $r_c(0_1^+)$, the charge radius of the proton $r_E^p = 0.84$ fm[51] is added in quadrature. For the NLEFT results, the first error bars are one standard deviation estimates due to stochastic errors and Euclidean time extrapolation. The second error bars are an estimate of systematic errors due to broken rotational invariance from the finite periodic volume and nonzero lattice spacing. Additional systematic errors due to the choice of interaction are described in the Supplementary Information.

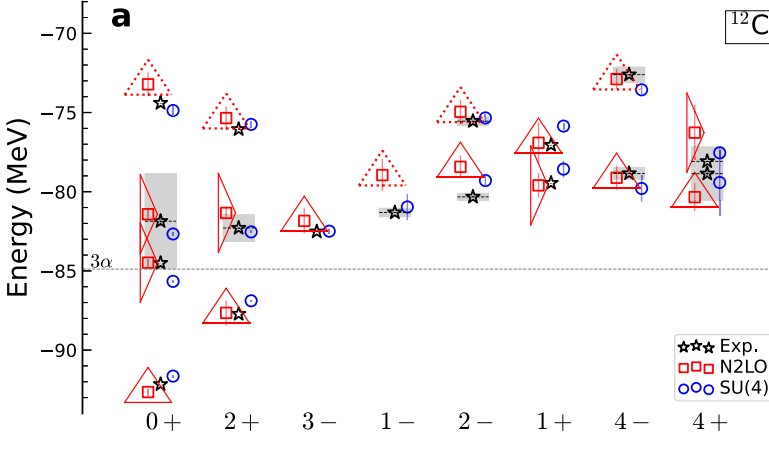

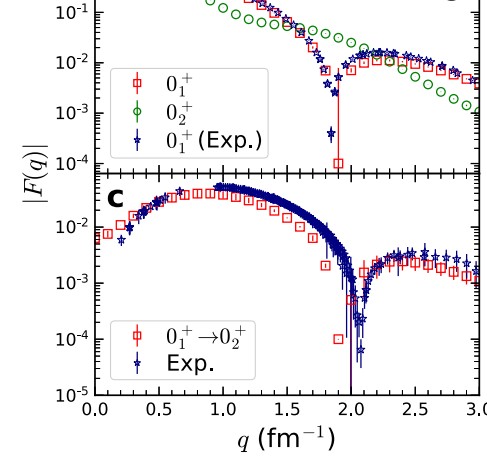

**Fig. 1 | Spectrum of $^{12}$C and charge form factors. a** Spectrum of $^{12}$C using N2LO interaction (red squares) and SU(4) interaction (blue circles) in comparison with experimental data (black stars). The error bars correspond to one standard deviation errors. The gray shaded regions indicate decay widths for cases where it has been measured. The triangular shapes indicate the intrinsic shape of each nuclear state, either equilateral or obtuse triangle arrangements of alpha clusters. The dotted lines for some equilateral triangles indicate significant distortions or large-amplitude displacements of the alpha clusters. **b, c** Absolute value of the charge form factor $F(q)$ using the SU(4) interaction. **b** The ground state (red squares) and Hoyle state (green circles), and **c** the transition from the ground state to the Hoyle state (red squares). The error bars correspond to one standard deviation errors. Experimental data (purple stars) are shown for comparison[5,38,40].

$2_1^+$, (d) $2_2^+$, (e) $3_1^-$, and (f) $0_3^+$ states. The black solid line at $\theta_2 = 180° − \theta_1$ separates the physical region (lower left) and the unphysical region (upper right). The dashed white triangle formed by the line segments $\theta_1 = 90°$, $\theta_2 = 90°$, and $\theta_2 = 90° − \theta_1$, represents cluster configurations that are right triangles. The interior region of the dashed white triangle corresponds to configurations that are acute triangles, and the exterior region corresponds to obtuse triangles. The other three white dashed line segments along the lines $\theta_1 = \theta_2$, $\theta_1 = \theta_3$, and $\theta_2 = \theta_3$ represent cluster configurations that are obtuse isosceles triangles. For the $0_1^+$ ground state, the probability distribution is strongly centered around an equilateral triangle, $\theta_1 = \theta_2 = \theta_3 = 60°$. The $2_1^+$ and $3_1^-$ states have similar equilateral triangular shapes. In contrast, the $0_2^+$ Hoyle state corresponds to an obtuse isosceles triangle. This finding is consistent with older NLEFT studies[33,34]. The $2_2^+$ and $0_3^+$ states also have obtuse isosceles triangular shapes.

We now define a model-independent two-dimensional projection of the nuclear density for the states of $^{12}$C. In order to construct this projection, we first identify the $x$ axis as the direction with the smallest

### Table 2 | Quadrupole moments and transition rates of $^{12}$C

|  | NLEFT | FMD | α cluster | NCSM | GCM | Exp. |
|---|---|---|---|---|---|---|
| $Q(2_1^+)$ | 6.8(3) (1.2) | – | – | 6.3 (3) | – | 8.1 (2.3) |
| $Q(2_2^+)$ | −35(1) (1) | – | – | – | – | – |
| $M(E0, 0_1^+ \rightarrow 0_2^+)$ | 4.8 (3) | 6.5 | 6.5 | – | 6.2 | 5.4 (2) |
| $M(E0, 0_1^+ \rightarrow 0_3^+)$ | 0.4 (3) | – | – | – | 3.6 | – |
| $M(E0, 0_2^+ \rightarrow 0_3^+)$ | 7.4 (4) | – | – | – | 47.0 | – |
| $B(E2, 2_1^+ \rightarrow 0_1^+)$ | 11.4(1) (4.3) | 8.7 | 9.2 | 8.7 (9) | – | 7.9 (4) |
| $B(E2, 2_1^+ \rightarrow 0_2^+)$ | 2.4(2) (7) | 3.8 | 0.8 | – | – | 2.6 (4) |

The quadrupole moments and transition rates of $^{12}$C are calculated using NLEFT with the SU(4) interaction and compared to calculations based on FMD[26], α cluster models[26], in-medium no-core shell model (NCSM)[42], generator coordinate method (GCM) calculation[46] and experiment[43,44]. Units for $Q$ and $M(E0)$ are $e\,\mathrm{fm}^2$, and for $B(E2)e^2\,\mathrm{fm}^4$. For the NLEFT results, the first error bars refer to the Euclidean time extrapolation uncertainties. The second error bars are an estimate of errors due to broken rotational invariance from the finite periodic volume and nonzero lattice spacing. Additional systematic errors due to the choice of interaction are described in the Supplementary Information.

RMS deviation of the nucleon positions relative to the center of mass. For the nuclear states that we have already identified as having an equilateral triangular shape, we rotate the density configurations along the $x$ axis so that one of the three clusters is pointing along the positive $z$ direction. We then symmetrize with respect to 0°, 120° and 240° rotations about the $x$ axis. For nuclear states that we have already identified as having an obtuse isosceles shape, we identify the $z$ axis as the direction with the longest RMS deviation of the nucleon positions relative to the CM. We then rotate the density configurations along the $z$ axis so that the alpha cluster closest to the CM lies on the positive $y$ axis.

In the right panel of Fig. 2, we show the density distribution of selected states of $^{12}$C for the SU(4) interaction. The density distributions obtained using the N2LO chiral interaction are presented in Fig. 3 and the Supplementary Information and are in excellent agreement with the SU(4) interaction results. The $0_1^+$, $2_1^+$, $3_1^-$, $4_1^-$, and $4_1^+$ states (see Supplementary Information for the last two) have similar intrinsic equilateral triangular shapes, consistent with an interpretation as members of a rotational band built on top of the $0_1^+$ state. The $0_2^+$, $2_2^+$, $4_2^+$ states (see Supplementary Information for the last one) have similar intrinsic obtuse isosceles triangle shapes and are consistent with belonging to a rotational band built on top of the $0_2^+$ state. These findings are consistent with previous studies in the literature based on group theoretical considerations[9]. We note that models where the Hoyle state has an equilateral triangle symmetry predict an additional $3^-$ and $4^-$ state in the Hoyle state rotational band.

The $0_3^+$ state has been discussed as a breathing mode excitation of the Hoyle state[13,45,46], but its detailed structure remains a matter of debate. For example, in a recent work the $0_3^+$ and Hoyle states are suggested to have an equilateral triangular shape[13,45]. A gas-like structure with a very large radius has also been predicted for the $0_3^+$ state[46]. Our lattice findings suggest that the $0_3^+$ state is a small-amplitude vibrational excitation of the Hoyle state. Our findings for the intrinsic shapes of the low-lying states of $^{12}$C are summarized by the triangular shapes in the left panel of Fig. 1. The triangular symbols indicate the intrinsic shape of each nuclear state, either equilateral or obtuse triangle arrangements of alpha clusters. The dotted lines for some

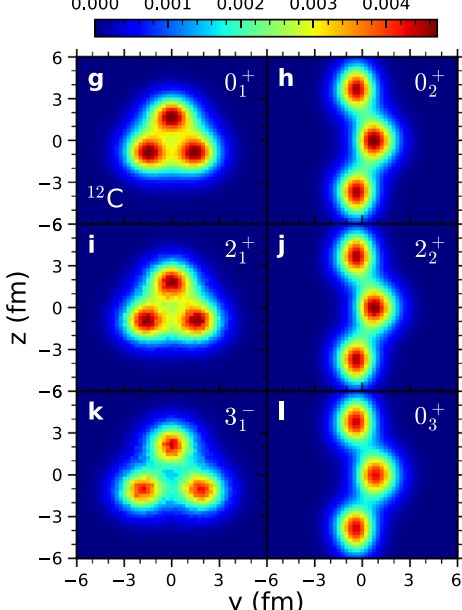

**Fig. 2 | Nuclear density distributions for several $^{12}$C states using the SU(4) interaction. a–f** Results for the density distribution of the two inner angles of the triangle formed by the three alpha clusters for the $0_1^+$, $0_2^+$, $2_1^+$, $2_2^+$, $3_1^-$, $0_3^+$ states respectively. The two axes are for the two inner angles $\theta_1$ and $\theta_2$ measured in degrees. **g–l** Results for the two-dimensional projection of the nuclear density for the $0_1^+$, $0_2^+$, $2_1^+$, $2_2^+$, $3_1^-$, $0_3^+$ states respectively. In each case the orientation of the shortest root-mean-square direction is aligned with the $x$ axis.

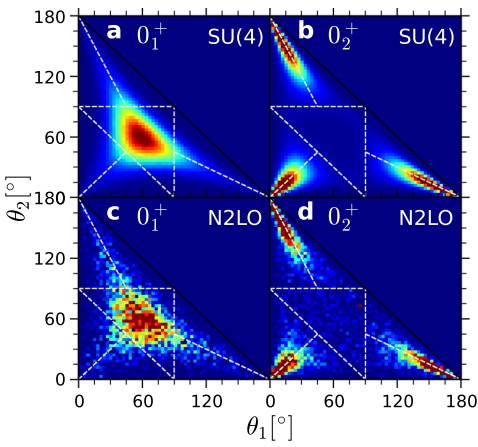
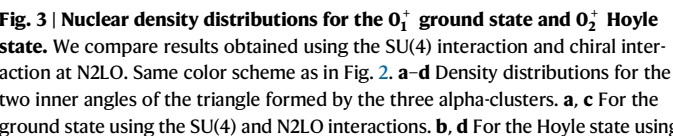
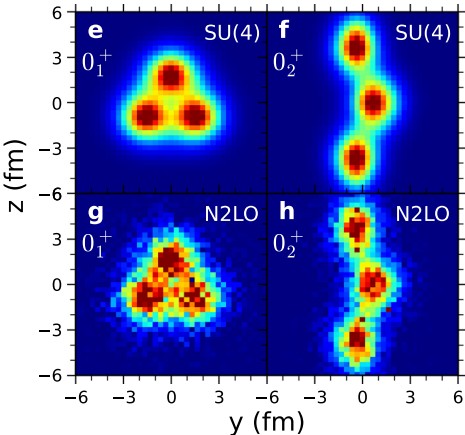

**Fig. 3 | Nuclear density distributions for the $0_1^+$ ground state and $0_2^+$ Hoyle state.** We compare results obtained using the SU(4) interaction and chiral interaction at N2LO. Same color scheme as in Fig. 2. **a–d** Density distributions for the two inner angles of the triangle formed by the three alpha-clusters. **a, c** For the ground state using the SU(4) and N2LO interactions. **b, d** For the Hoyle state using the SU(4) and N2LO interactions. The two axes are for the two inner angles $\theta_1$ and $\theta_2$ measured in degrees. **e–h** Two-dimensional projection of the nuclear density. **e, g** For the ground state using the SU(4) and N2LO interactions. **f, h** For the Hoyle state using the SU(4) and N2LO interactions.

equilateral triangles indicate significant distortions or large-amplitude displacements of the alpha clusters. Examples of these states are shown in the Supplementary Information. We find that all of the low-lying states with an equilateral triangle formation have significant overlap with an initial state composed of an antisymmetrized product of mean-field shell model states. This constitutes the aforementioned duality between shell model and cluster states. In contrast, all of the low-lying states with an obtuse isosceles triangle formation have very little overlap with shell model initial states.

In Fig. 3 we show that the SU(4) interaction and the N2LO chiral interactions produce the same nuclear structures. We compare SU(4) and N2LO chiral results for the $0_1^+$ ground state and $0_2^+$ Hoyle state. The N2LO chiral interaction calculations require an order of magnitude more computational effort due to Monte Carlo sign oscillations. The resulting greater statistical noise for the N2LO chiral results should not be misinterpreted as a difference in nuclear structure. To facilitate the calculation for N2LO, the comparison is performed at smaller projection time and smaller resolution for the densities.

We note that for the obtuse triangular configurations, the distribution of obtuse angles extends all the way to 180°. This suggests that the intrinsic shape is not a rigid triangular shape with fixed angles, but rather a superposition of obtuse triangular shapes spanning a range of angles. The preference for the obtuse triangular shape can be understood as arising from the need to satisfy wave function orthogonality with respect to the other nuclear states favoring the predominantly equilateral triangle configuration.

Previous work using Green's function Monte Carlo has also found signals of the Hoyle state, with a slightly higher excitation energy[21]. The density of the ground state and the transition between the Hoyle state and ground state is nicely reproduced. Their conclusion concerning intrinsic shapes is in line with the results presented here, that the ground state is dominated by an approximately equilateral distribution of alphas while the Hoyle state has an approximately linear distribution[21]. In a recent publication[20], the shape of Hoyle state has also been studied in the framework of Monte Carlo shell model, and the intrinsic density is defined in terms a Q-aligned state where each basis state in the many-body wave function is rotated according to their principal axes. In that work, conclusions similar to the results presented here were obtained regarding the structure of the ground state, Hoyle state, and $0_3^+$ states, though there are some differences in the details. A detailed quantitative comparison of intrinsic shapes are unfortunately not possible at present.

In summary, we have presented a model-independent density projection of the geometry of the nuclear states of $^{12}$C using nuclear lattice effective field theory. We find excellent agreement using two different interactions, an SU(4) interaction and an ab initio N2LO chiral interaction. We find that the Hoyle state and its $2_2^+$ and $4_2^+$ rotational excitations are composed of an obtuse isosceles triangular arrangement of alpha clusters. All of the low-lying nuclear states of $^{12}$C have an intrinsic shape composed of three alpha clusters forming either an equilateral triangle or an obtuse triangle. From these basic structural formations, the various nuclear states correspond to different rotational and vibrational excitations as well as either distortions or large-amplitude displacements of the alpha clusters. Future studies are planned to revisit this analysis using high-fidelity chiral effective field theory interactions at higher orders and to compute decay widths of resonance states using Euclidean time response functions.

## Methods

For the lattice simulations presented here, we use a spatial lattice spacing $a = 1.64$ fm and a temporal lattice spacing of $a_t = 0.55$ fm/$c$, where $c$ is the speed of light. While the individual nucleons must reside on lattice sites, the center of mass (CM) of the $^{12}$C nucleus is constrained to a much finer three-dimensional grid of lattice spacing $a/12 = 0.137$ fm, which equals the resolution of our density projection for each $^{12}$C state. resolution scale of 0.137 fm. We use two different lattice interactions for this study. The first is the SU(4) interaction between the nucleons that is independent of spin and isospin. It has the form

$$V = \frac{C_2}{2!}\sum_{\mathbf{n}} \tilde{\rho}(\mathbf{n})^2 + \frac{C_3}{3!}\sum_{\mathbf{n}} \tilde{\rho}(\mathbf{n})^3, \tag{1}$$

where $C_2$ and $C_3$ are the two-body and the three-body interaction coefficients, respectively. The vector $\mathbf{n}$ denotes the spatial lattice sites. The definition of the smeared density operator $\tilde{\rho}(\mathbf{n})$ is given in Supplementary Information, and it entails two parameters, $s_L$, and $s_{NL}$. The four parameters $C_2$, $C_3$, $s_L$, and $s_{NL}$ are determined by a joint fit to the ground-state energies of $^4$He and $^{12}$C, to the ground-state charge radius of $^{12}$C, and to several electromagnetic transition rates. The same type of lattice interaction has been used to describe the ground state energies of light and medium-mass nuclei[47], the thermodynamics of symmetric nuclear matter[48], and it well reproduces the low-energy spectrum of $^{12}$C[49].

As the calculations using the SU(4) interaction can be done efficiently with high statistics, most of the results presented in this work are obtained using this interaction. However, we also validate key results by performing a second set calculations using an ab initio chiral effective field theory interaction at N2LO. As the N2LO chiral interaction has many more interactions and produces sign oscillations in the Monte Carlo simulations, it requires an order-of-magnitude greater computational resources. The chiral interaction up to N2LO is defined as

$$V_{2N} = V_{SU(4)} + V_{OPE} + V_{contact}^{(Q/\Lambda_\chi)^0} + V_{contact}^{(Q/\Lambda_\chi)^2} + V_{3N}, \quad (2)$$

where $Q$ is the low-energy scale (external momentum or pion mass), $\Lambda_\chi \simeq 700$ MeV is the hard scale, $V_{OPE}$ is the one-pion-exchange potential, and $V_{3N}$ gives the leading three-nucleon force that appears at N2LO. A full description of the N2LO chiral interaction is given in the Supplementary Information.

We use an assortment of different initial states for each state of $^{12}$C and verify that our choice of initial state does not affect the final observables. The initial states we consider include a wide variety of mean-field states composed of products of harmonic oscillator (shell model) states as well as different geometric configurations of alpha clusters. We use the pinhole algorithm to determine the probability distribution for the nucleon positions, spins, and isospins[50]. For each pinhole configuration, we know the positions of all $A$ nucleons, and thus the position of each nucleon relative to the center of mass is easily calculated. From this information, we can compute any observable that does not involve displacements of the nucleons. In the Supplementary Information, we give details of the auxiliary-field lattice Monte Carlo calculations, pinhole algorithm, nucleon density operators, Euclidean time extrapolation, SU(4) interaction parameters and systematic errors, N2LO chiral interaction parameters and results, electromagnetic observables, density projection of lattice states, and model-independent probes of cluster geometry.

## Data availability
All of the data generated in this study are available at this public repository folder.

## Code availability
All of the codes produced in association with this work have been stored and can be obtained upon request from the authors, subject to possible export control constraints.

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

## Acknowledgements

We are grateful for discussions with members of the Nuclear Lattice Effective Field Theory Collaboration as well as Scott Bogner, Jerry Draayer, Martin Freer, Heiko Hergert, Morten Hjorth-Jensen, and Witek Nazarewicz. We acknowledge funding by the Deutsche Forschungsgemeinschaft (DFG, German Research Foundation) and the NSFC through the funds provided to the Sino-German Collaborative Research Center TRR110 "Symmetries and the Emergence of Structure in QCD" (DFG Project ID 196253076 - TRR 110, NSFC Grant No. 12070131001), the Chinese Academy of Sciences (CAS) President's International Fellowship Initiative (PIFI) (Grant No. 2018DM0034), the NSAF (Grant No. U1930403), the National Natural Science Foundation of China (Grant No. 12275259), Volkswagen Stiftung (Grant No. 93562), the European Research Council (ERC) under the European Union's Horizon 2020 research and innovation programme (ERC AdG EXOTIC, grant agreement No. 101018170) and the U.S. Department of Energy (DE-SC0013365 and DE-SC0021152) and the Nuclear Computational Low-Energy Initiative (NUCLEI) SciDAC-4 project (DE-SC0018083) as well as computational resources provided by the Gauss Centre for Supercomputing e.V. (www.gauss-centre.eu) for computing time on the GCS Supercomputer JUWELS at Jülich Supercomputing Centre (JSC) and special GPU time allocated on JURECA-DC as well as the Oak Ridge Leadership Computing Facility through the INCITE award "Ab-initio nuclear structure and nuclear reactions".

## Author contributions

S.S. performed the theoretical work, adapted codes developed by the NLEFT collaboration, and performed the required lattice MC simulations. S.E. and B.-N.L. helped to develop and perform the N2LO chiral EFT calculations. T.L., D.L., and U.-G.M. supervised the direction of the research. All authors were involved in the writing, editing, and review of this work.

## Competing interests

The authors declare no competing interests.
