## [Peer Review File · Nature Communications]

Emergent geometry and duality in the carbon nucleusEditorial Note: Parts of this Peer Review File have been redacted as indicated to remove third-party material where no permission to publish could be obtained.

REVIEWER COMMENTS

Reviewer #1 (Remarks to the Author):

The authors have applied the state-of-the-art ab initio framework of nuclear lattice effective field theory to describe the structure of the ^{12}C nucleus. Furthermore, the first model independent tomographic scan of the three dimensional geometry of the individual states is introduced and all low-lying states in ^{12}C have been attributed either an equilateral triangle or a obtuse triangle shape with alpha clusters located at the vertices of triangle. The calculated energies, quadrupole moments, electromagnetic transition rates, charge densities and form factors of the low-lying states in ^{12}C are in a very good agreement with the experimental data. To the best of my knowledge, this is a first calculation that provides detailed information on the three-dimensional geometry of the low-lying states in the ^{12}C nucleus within the ab initio framework. In particular, the authors have shown that the Hoyle state and rotational excitations built on top of the Hoyle state display obtuse isoscales triangular configuration of alpha clusters, while the ground-state and the corresponding rotational excitations display equilateral triangular configuration of alpha clusters.

The methodology used in this study is state-of-the-art and therefore it meets every standard in the field of theoretical nuclear physics. The section describing the methods is quite detailed and contains all numerical details necessary to reproduce the results.

The authors have performed an analysis of possible systematic errors in their calculations (e.g. finite Euclidean projection time), however by looking at Fig. 1 it seems that the lattice spacing influences the results considerably and in some cases with same order of magnitude as the three-body force. Perhaps the authors could include some additional comment on this source of systematic error.

To conclude, I believe that proposed manuscript presents a state of the art study of low-lying states in the ^{12}C isotope and I would certainly recommend publication in Nature Communications.

Reviewer #2 (Remarks to the Author):

Report on " Emergent geometry and duality in the carbon nucleus", Shen et al.

This manuscript presents an important contribution to further understanding the structure of one of the key light nuclei whose properties are a test of our understanding of the interplay of fundamental nuclear forces and the structure of nuclei. The framework used is lattice nuclear effective field theory, which is probably the best basis for the testing of these ideas. What emerges from the present set of calculations, which builds on earlier work by this group, is the most comprehensive description of the structure of the states of ^{12}C in terms of the geometric properties. This really is a big step forward which consolidates many other works using different models which have produced conclusions aligned with the present. Given the fundamental nature of the present approach and the very comprehensive characterisation of the states in ^{12}C I believe this work will be the foundation for the field going forward and is worthy of publication in the present journal. I do have a number of questions/comments which are listed below, which are aimed at improving the clarity etc....

1. I struggled to understand the terminology, topographic projection. For my taste the density plots which are shown on the RHS of figure 2 are in practice no different to those which are associated with AMD or FMD type calculations. One is left with an impression that the term topographic projection is rather introduced to give a superficial appearance of novelty. If this is a misunderstanding then I think the terminology needs a much better discussion/presentation or toning down.

2. Page 1: "In our calculations, we use a simple interaction between the nucleons that is independent of spin and whether the nucleon is a proton or neutron, i.e., of isospin" This is a pretty big assumption and one is left wondering how robust the conclusions are given the assumption. Some confidence

might be given to the reader in either direction.

3. Figure 1 (LHS): How to be sure calculations are converged at the lattice spacing used?

4. Figure 1 (RHS): Missing the minimum in the distribution (b) indicates that the size of the ground or Hoyle state is not right in the calculations; how big is the effect.

5. Figure 1 (LHS). Is there any perspective the calculations can offer on widths, e.g. 0+ and 4+ states at similar energies but with different widths. Clearly the widths of the states offer a key signature of the structure. If the conclusions here are robust then the widths should be reproduced.

6. Page 3/4: "we conclude that our process of identifying alpha cluster configurations is accurate and free of significant artifacts." – is there a unique solution, or more than one?

7. Figure 2 and elsewhere: How can one understand why for the non-equilateral configurations that a particular, and common, obtuse angle is arrived at and why is that arrangement stable. Give the reflection asymmetry what are the consequences for parity?

8. A recent publication, T. Otsuka, T. Abe, T. Yoshida, Y. Tsunoda, N. Shimizu, N. Itagaki, Y. Utsuno, J. Vary, P. Maris & H. Ueno, Nature Communications volume 13, Article number: 2234 (2022), comes to overlapping conclusions with the present work and a comment by the authors on similarity differences and hence the robustness of the emergent understanding would be welcome.

Reviewer #3 (Remarks to the Author):

Review Report

Title: Emergent geometry and duality in the carbon nucleus

Authors: S. Shen, T.A. Lahde, D. Lee and U.G. Meissner,

This manuscript shows theoretical results of a lattice simulation of multi-nucleon systems applied to the structure study of carbon-12 nucleus. The obtained numerical results are compared to experimental data, for which the authors claim that the agreement is good. The density distributions of nucleons are discussed, and the configurations of three alpha clusters are mentioned.

We first observe that the theoretical model adopted in this work is extremely simplified, and cannot be a good approximation. Despite protons and neutrons are fermions, their spins are ignored. Protons and neutrons are not distinguished, or the isospin is totally absent. In nuclear forces, however, the spin-isospin channel is particularly important, while this feature is completely ignored in the present work. The lattice calculations were performed, but it has no relation to the studies with ab initio aspects published in PRL a few years ago by the same group (some authors are different). The parameters of the model should have been adjusted to the experimental data to compare with. Otherwise, there is no way to fix their values. Thus, the alleged agreement does not provide with evidences of the reliability of the model.

The authors claim that the present simple model has common features with the AMD or BEC approaches. I doubt this statement. The calculations by the AMD and BEC have been performed with more reasonable and realistic interactions. This claim must be dropped off.

The final conclusion includes the configurations of the alpha clusters. The way to extract the density profile is not clearly presented in the text. Moreover, the density distributions shown in the present article significantly differs from the corresponding ones obtained by first-principles calculations reported in Nature Commun. 13, 2234 (2022), where the ground-state density profile does not show an equilateral triangle, or the Hoyle state shows a much smaller angle among the three clusters. Considering the fact that the work in Nature Commun. 13, 2234 (2022) is first-principles calculations, the present work, which is empirical, is supposed to reproduce the main features of the Nature Commun. paper in addition to the fitted experimental data. As the final results fail to do so, the basic model assumption and/or the overall many-body methodology very likely contains flaw.

The usage of unusual terminologies such as tomography does not help the understanding of the general readers, and is better avoided. In addition, the expression like "first model-independent tomographic scan" is very much misleading, and should be removed.

Some relevant papers are not cited.

Considering these findings, I cannot recommend the publication of this article in Nature Communications. The publications in more specialized journals can still be difficult, I am afraid. It is advised that if submitted to another journal, in the extraction of the density profiles discussed in Methods, the relations to the work by the GFMC approach and to the Nature Commun. Paper mentioned above must be clearly stated in the revised manuscript. As a matter of fact, strong similarities exist, although these two approaches are realistic and *ab initio*.

We thank all three referees for the positive comments, useful suggestions, and constructive critiques. We have taken all of the suggestions and comments seriously and spent the past three months performing new calculations to address all of the concerns. We have also added two additional authors (Serdar Elhatisari and Bing-Nan Lu) to help with the new calculations and the many revisions to the paper.

Reply to Reviewer 1

We thank the referee for the positive review and the very helpful suggestion for improvement. We have implemented the suggestion.

The authors have performed an analysis of possible systematic errors in their calculations (e.g. finite Euclidean projection time), however by looking at Fig. 1 it seems that the lattice spacing influences the results considerably and in some cases with same order of magnitude as the three-body force. Perhaps the authors could include some additional comment on this source of systematic error.

We thank the referee for this suggestion. The differences between the $a = 1.97$ fm results and $a = 1.64$ fm results from Eur. Phys. J. A 57, 276 (2021) can be explained by the significantly larger lattice errors at $a = 1.97$ fm. As shown in Phys. Rev. D 92, 014506 (2015), the lattice artifacts become significantly larger at lattice spacing $a > 1.7$ fm for calculations of ^8Be . In the revised figure we no longer show results at $a = 1.97$ fm lattice spacing. Instead we present calculations performed at $a = 1.64$ fm using the SU(4) interaction and chiral interactions at next-to-next-to-leading order (N²LO).

In the section “SU(4) interaction parameters and systematic errors” of Methods we have added a discussion concerning this source of error:

◇ In a previous study of the ^{12}C spectrum using NLEFT,¹ two different lattice spacings were used. The differences between the $a = 1.97$ fm results and $a = 1.64$ fm results can be explained by the significantly larger lattice errors at $a = 1.97$ fm. As shown in Ref.² for calculations of ^8Be , the lattice artifacts become significantly larger for $a > 1.7$ fm. This explains the choice of lattice spacing $a = 1.64$ fm for this analysis. Work currently in progress using a smaller lattice spacing of $a = 1.32$ fm provides confirmation that the lattice artifacts at $a = 1.64$ fm are less than 1 or 2 MeV in binding energy for the spectrum of ^{12}C . This is reflected in the excellent agreement with the observed binding energies. ◇

Reply to Reviewer 2

We thank referee for the positive comments and the many useful suggestions. We have implemented all of the suggestions.

1. I struggled to understand the terminology, topographic projection. For my taste the density plots which are shown on the RHS of figure 2 are in practice no different to those which are associated with AMD or FMD type calculations. One is left with an impression that the term topographic projection is rather introduced to give a superficial appearance of novelty. If this is a misunderstanding then I think the terminology needs a much better discussion/presentation or toning down.

For the AMD or FMD calculations, the density is plotted by showing the contribution of various basis states comprising the wave function and their relative importance. This definition is dependent on the choice of basis. In this work, the density is defined in a model-independent fashion in terms of the full A -body probability distribution. But we do agree that the “tomography” terminology is not necessary, and so in the revised text we have changed it to “two-dimensional projection” and “density projection”.

2. Page 1: “In our calculations, we use a simple interaction between the nucleons that is independent of spin and whether the nucleon is a proton or neutron, i.e., of isospin” This is a pretty big assumption

and one is left wondering how robust the conclusions are given the assumption. Some confidence might be given to the reader in either direction.

To address this issue, we have done additional calculations using ab initio chiral interactions at next-to-next-to leading order (N2LO). The revised Fig. 1 shows the comparison of binding energies between the SU(4) results and the N2LO chiral results. In all cases the agreement is very good. In Methods, we present the details of the N2LO chiral interaction in the section “N2LO chiral interaction parameters and results”. A comparison of the two-dimensional density projections obtained using the SU(4) interaction and the N2LO chiral interaction are shown. In all cases the agreement is excellent.

3. *Figure 1 (LHS): How to be sure calculations are converged at the lattice spacing used?*

In the section “SU(4) interaction parameters and systematic errors” of Methods we have added a discussion concerning lattice discretization errors:

◇ In a previous study of ^{12}C spectrum using NLEFT,¹ two different lattice spacings were used. The differences between the $a = 1.97$ fm results and $a = 1.64$ fm results can be explained by the significantly larger lattice errors at $a = 1.97$ fm. As shown in Ref.² for calculations of ^8Be , the lattice artifacts become significantly larger for $a > 1.7$ fm. This explains the choice of lattice spacing $a = 1.64$ fm for this analysis. Work currently in progress using a smaller lattice spacing of $a = 1.32$ fm provides confirmation that the lattice artifacts at $a = 1.64$ fm are less than 1 or 2 MeV in binding energy for the spectrum of ^{12}C . This is reflected in the excellent agreement with the observed binding energies. ◇

4. *Figure 1 (RHS): Missing the minimum in the distribution (b) indicates that the size of the ground or Hoyle state is not right in the calculations; how big is the effect.*

We thank the referee for this observation. The discrepancy can be seen more clearly in Fig. S12 of Methods. As one can see from Panel (a) of Fig. S12, the charge density of the ground state is very well reproduced. The discrepancy is apparent in Panel (b) of Fig. S12 and comes from the tail of the Hoyle state at a radius of about 4 fm. This small difference in the tail of the Hoyle state has little effect on the overall structure of the Hoyle state. However it has an impact on electromagnetic transitions between the Hoyle state and the members of the ground state rotational band. This is studied in detail in Table 3 and Fig. S2 of Methods.

In the revised section “SU(4) interaction parameters and systematic errors” in Methods we write:

◇ In Fig. S2, we show the ^{12}C form factors obtained at finite Euclidean projection time $t = 0.4 \text{ MeV}^{-1}$ for the three interactions V1, V2, and V3, characterized by a different strength of the non-local smearing. As s_L is varied, the value of s_{NL} is adjusted so that ground state radius is kept constant. This results in the ground state form factor remaining almost the same for the three different interactions. There is also not a significant change to the overall charge density of the Hoyle. However, the transition from the ground state to the Hoyle state is more sensitive to changes in s_L . The shapes and positions of the maxima and minima of the transition form factor have some dependence on s_L , and this is also reflected in significant changes to the electromagnetic transitions between the Hoyle state and members of the ground state rotational band. ◇

5. *Figure 1 (LHS). Is there any perspective the calculations can offer on widths, e.g. $0+$ and $4+$ states at similar energies but with different widths. Clearly the widths of the states offer a key signature of the structure. If the conclusions here are robust then the widths should be reproduced.*

We thank the referee for this suggestion. The calculation of resonance widths using Monte Carlo methods is an important but very difficult challenge. It requires the calculation of Euclidean response functions and then using approximate inverse Laplace transform methods to extract spectral functions.

Such calculations are beyond the scope of the current work. However, the collaboration is actively working on developing this new technology to make accurate and efficient calculations of resonance widths. At the end of the main text, we now write:

◇ Future studies are planned to revisit this analysis using high-fidelity chiral effective field theory interactions at higher orders and to compute decay widths of resonance states using Euclidean time response functions. ◇

6. Page 3/4: “we conclude that our process of identifying alpha cluster configurations is accurate and free of significant artifacts.” – is there a unique solution, or more than one?

We thank the referee for asking this interesting and important question. We have sharpened the language in the revised draft. We now write:

◇ The root-mean-square (RMS) matter radii of the alpha clusters defined in this manner range from 1.65 fm to 1.71 fm for the low-lying states in ^{12}C . This is very close to the matter radius of an isolated alpha particle using the same interactions, $r_\alpha = 1.63$ fm, and this gives us confidence that we are indeed identifying alpha clusters. While the process of identifying alpha clusters is not unique, the method we are using here is defined unambiguously from the twelve-nucleon spatial probability distribution. So long as we are not probing very short-distance correlations between nucleons, the twelve-nucleon spatial probability distribution is model independent. Therefore, our definition of the alpha cluster geometry is also model independent. ◇

7. Figure 2 and elsewhere: How can one understand why for the non-equilateral configurations that a particular, and common, obtuse angle is arrived at and why is that arrangement stable. Give the reflection asymmetry what are the consequences for parity?

In the revised text, we give a clearer description of the definition of our model-independent two-dimensional projection:

◇ We now define a model-independent two-dimensional projection of the nuclear density for the states of ^{12}C . In order to construct this projection, we first identify the x axis as the direction with the smallest RMS deviation of the nucleon positions relative to the center of mass. For the nuclear states that we have already identified as having an equilateral triangular shape, we rotate the density configurations along the x axis so that one of the three clusters is pointing along the positive z direction. We then symmetrize with respect to 0° , 120° and 240° rotations about the x axis. For nuclear states that we have already identified as having an obtuse isosceles shape, we identify the z axis as the direction with the longest RMS deviation of the nucleon positions relative to the CM. We then rotate the density configurations along the z axis so that the alpha cluster closest to the CM lies on the positive y axis. ◇

While the intrinsic shape given by the projected two-dimensional density breaks parity, the nuclear wave function themselves do not break parity. From the description above it is clear that the oriented alignment of the intrinsic shape in a particular manner produces this difference.

In the section “N2LO chiral interaction parameters and results” in Methods, we now write:

◇ We note that for the obtuse triangular configurations, the distribution of obtuse angles extends all the way to 180° . This suggests that the intrinsic shape is not a rigid triangular shape with fixed angles, but rather a superposition of obtuse triangular shapes spanning a range of angles. The preference for the obtuse triangular shape can be understood as arising from the need to satisfy wave function orthogonality with respect to the other nuclear states favoring the predominantly equilateral triangle configuration. ◇

8. A recent publication, T. Otsuka, T. Abe, T. Yoshida, Y. Tsunoda, N. Shimizu, N. Itagaki, Y. Utsuno, J. Vary, P. Maris, and H. Ueno, *Nature Communications* volume 13, Article number: 2234 (2022), comes to

overlapping conclusions with the present work and a comment by the authors on similarity differences and hence the robustness of the emergent understanding would be welcome.

We thank the referee for this suggestion. This very nice reference became publicly available while our publication was being refereed. In the revised text, we now write:

◇ In a recent publication,⁴ the shape of Hoyle state has also been studied in the framework of a Monte-Carlo shell model, and the intrinsic density is defined in terms a Q -aligned state where each basis state in the many-body wave function is rotated according to their principal axes. In that work, conclusions similar to the results presented here were obtained regarding the structure of the ground state, Hoyle state, and 0_3^+ states, though there are some differences in the details. A detailed quantitative comparison of intrinsic shapes are unfortunately not possible because the Q -aligned state definition depends on the specific choice of many-body basis states. ◇

Reply to Reviewer 3

We thank the referee for the careful reading of our work and for the constructive criticism. We took the criticism very seriously and used it as motivation to produce a much stronger paper than the original manuscript. We added two additional authors, Serdar Elhatisari and Bing-Nan Lu, who provided additional expertise needed to perform the new *ab initio* calculations at N2LO order in chiral effective field theory. The author team spent three full months of intensive work writing the new N2LO codes and performing the new calculations. These calculations are an order of magnitude more computationally intensive than the original calculations using the SU(4) interaction, and we spent nearly all of the remaining supercomputing time available to complete this work.

We first observe that the theoretical model adopted in this work is extremely simplified, and cannot be a good approximation. Despite protons and neutrons are fermions, their spins are ignored. Protons and neutrons are not distinguished, or the isospin is totally absent. In nuclear forces, however, the spin-isospin channel is particularly important, while this feature is completely ignored in the present work.

The simplified nature of the SU(4) interaction is a fair criticism of the original manuscript. Therefore we have performed new *ab initio* calculations using chiral effective field theory at N2LO. The agreement between the SU(4) interaction results and the N2LO chiral interaction results are excellent. The agreement for the binding energies is shown in Fig. 1, and the agreement for the nuclear structure and intrinsic shapes are detailed in the section “N2LO chiral interaction parameters and results” in Methods. In the concluding paragraph of the revised main text, we write:

◇ We find excellent agreement using two different interactions, a simplified SU(4) interaction and an *ab initio* N2LO chiral interaction. ◇

The lattice calculations were performed, but it has no relation to the studies with ab initio aspects published in PRL a few years ago by the same group (some authors are different). The parameters of the model should have been adjusted to the experimental data to compare with. Otherwise, there is no way to fix their values. Thus, the alleged agreement does not provide with evidences of the reliability of the model.

This is a fair criticism. The SU(4) interaction does not qualify as an *ab initio* interaction. As noted above, we have therefore performed new *ab initio* calculations using chiral effective field theory at N2LO. The agreement between the SU(4) interaction results and the N2LO chiral interaction results are excellent. The agreement for the binding energies is shown in Fig. 1, and the agreement for the nuclear structure and intrinsic shapes are detailed in the section “N2LO chiral interaction parameters and results” in Methods.

The authors claim that the present simple model has common features with the AMD or BEC ap-

proaches. I doubt this statement. The calculations by the AMD and BEC have been performed with more reasonable and realistic interactions. This claim must be dropped off.

This is a fair criticism. We have removed the corresponding sentences from the revised manuscript.

The final conclusion includes the configurations of the alpha clusters. The way to extract the density profile is not clearly presented in the text. Moreover, the density distributions shown in the present article significantly differs from the corresponding ones obtained by first-principles calculations reported in Nature Commun. 13, 2234 (2022), where the ground-state density profile does not show an equilateral triangle, or the Hoyle state shows a much smaller angle among the three clusters. Considering the fact that the work in Nature Commun. 13, 2234 (2022) is first-principles calculations, the present work, which is empirical, is supposed to reproduce the main features of the Nature Commun. paper in addition to the fitted experimental data. As the final results fail to do so, the basic model assumption and/or the overall many-body methodology very likely contains flaw.

We agree with the referee that the explanation of the extraction of the density profile is not clearly presented in the text. In the revised text, we write:

◇ We now define a model-independent two-dimensional projection of the nuclear density for the states of ^{12}C . In order to construct this projection, we first identify the x axis as the direction with the smallest RMS deviation of the nucleon positions relative to the center of mass. For the nuclear states that we have already identified as having an equilateral triangular shape, we rotate the density configurations along the x axis so that one of the three clusters is pointing along the positive z direction. We then symmetrize with respect to 0° , 120° and 240° rotations about the x axis. For nuclear states that we have already identified as having an obtuse isosceles shape, we identify the z axis as the direction with the longest RMS deviation of the nucleon positions relative to the CM. We then rotate the density configurations along the z axis so that the alpha cluster closest to the CM lies on the positive y axis. ◇

The referee makes the argument that the results presented in our work are likely not correct because they do not agree fully with the intrinsic structures presented in Ref.⁴ In the following, we present a more likely explanation of the disagreement, and it does not require either work to be incorrect.

While Ref.⁴ is a very nice work, it never claims to compute intrinsic densities that are model independent and indeed it does not. In Ref.⁴ the intrinsic density is defined in terms a Q -aligned state where each basis state in the many-body wave function is rotated according to their principal axes. A detailed quantitative comparison of intrinsic shapes of our work and that of Ref.⁴ is unfortunately not possible because the Q -aligned state definition depends on the specific choice of many-body basis states.

This can be seen easily from a simple model of two particles with equal masses interacting in three dimensions via a two-body potential that is a quadratic function of the separation distance between particles. Let x, y, z be the coordinates for the relative separation vector between the two particles. The ground state of the system will have a three-dimensional wave function of the form

$$\psi(x, y, z) = \frac{1}{\pi^{3/4} R^{3/2}} \exp \left[-\frac{x^2}{2R^2} - \frac{y^2}{2R^2} - \frac{z^2}{2R^2} \right]. \quad (1)$$

Let us now perform the basis alignment procedure using two different sets of bases. The first basis we use is the position eigenstate basis. This is the basis we use in our calculations and is the standard basis for defining intrinsic densities and shapes. It is the standard basis as well as the most logical choice since all density operators are diagonal in this basis. There are no off-diagonal cross terms that arise when computing expectation values of density operators.

If we use the position space basis and align the largest principal axis along the z -axis, with no bias

distinguishing the $+z$ and $-z$ directions, we then get an intrinsic density that is extremely prolate,

$$\rho_{\text{position}}(x, y, z) = \frac{1}{\pi^{1/2}R} \exp\left[-\frac{z^2}{R^2}\right] \delta(x)\delta(y). \quad (2)$$

Suppose now we instead perform the basis alignment procedure using eigenstates of our two-body 3D harmonic oscillator. In that case we need only one basis state corresponding to our ground state wave function itself. The resulting intrinsic density is then spherical,

$$\rho_{\text{harmonic}}(x, y, z) = \frac{1}{\pi^{3/2}R^3} \exp\left[-\frac{x^2}{R^2} - \frac{y^2}{R^2} - \frac{z^2}{R^2}\right]. \quad (3)$$

Clearly the Q -aligned state definition does not allow for a model independent prediction of intrinsic densities and shapes.

In contrast, the definition of intrinsic densities and shapes in our work has a solid theoretical foundation and is model independent. Furthermore, the new calculations using *ab initio* N2LO chiral interactions provide strong confirmation of the accuracy of our results. We note also that the Green's function Monte Carlo calculations in Ref.⁵ also lends support to our conclusions:

“However, the Hoyle state density is peaked at $r = 0$ in both the VMC and GFMC calculations. A possible interpretation of these results is that the ground state is dominated by an approximately equilateral distribution of alphas while the Hoyle state has an approximately linear distribution.”

The usage of unusual terminologies such as tomography does not help the understanding of the general readers, and is better avoided. In addition, the expression like “first model-independent tomographic scan” is very much misleading, and should be removed.

We thank the referee for this suggestion. We agree that the “tomography” terminology is not necessary, and so in the revised text we have changed it to “two-dimensional projection” and “density projection”. Since the model independence of our analysis is a key attribute of our paper, we are keeping the “model independence” terminology.

Some relevant papers are not cited.

We thank the referee for this comment. In the revised text we now write:

◇ Previous work using Green's function Monte Carlo has also found signals of the Hoyle state, with a slightly higher excitation energy.⁵ The density of the ground state and the transition between the Hoyle state and ground state is nicely reproduced. Their conclusion concerning intrinsic shapes is in line with the results presented here, that the ground state is dominated by an approximately equilateral distribution of alphas while the Hoyle state has an approximately linear distribution.⁵

In a recent publication,⁴ the shape of Hoyle state has also been studied in the framework of Monte Carlo shell model, and the intrinsic density is defined in terms a Q -aligned state where each basis state in the many-body wave function is aligned according to principal axes. In that work, conclusions similar to the results presented here were obtained regarding the structure of the ground state, Hoyle state, and 0_3^+ states are reached, though there are some differences in the details. A detailed quantitative comparison is unfortunately not possible since the Q -aligned state definition depends on the specific choice of many-body basis states. ◇

References

1. S. Shen, T. A. Lähde, D. Lee and U.-G. Meißner, Eur. Phys. J. A **57**, no.9, 276 (2021).

2. B. N. Lu, T. A. Lähde, D. Lee and U.-G. Meißner, Phys. Lett. B **760**, 309-313 (2016).
3. N. Li, S. Elhatisari, E. Epelbaum, D. Lee, B.-N. Lu and U.-G. Meißner, Phys. Rev. C **98**, 044002 (2018).
4. T. Otsuka, *et al.*, Nat. Commun. 13 (2022) 1, 2234.
5. J. Carlson, *et al.*, Rev. Mod. Phys. **87**, 1067 (2015).

REVIEWER COMMENTS

Reviewer #1 (Remarks to the Author):

After reading the author response and the revised version of the manuscript I can confirm that the authors have modified the manuscript accordingly to my comments and suggestions. Therefore, I can recommend the paper for publication in Nature communications.

Reviewer #2 (Remarks to the Author):

I am happy that the authors have addressed questions I raised and comments that I made. I believe this work is of high importance and could be accepted for publication. The authors have worked hard to demonstrate the rigor of their approach

Reviewer #3 (Remarks to the Author):

Review Report

Title: Emergent geometry and duality in the carbon nucleus

Version: 2

Authors: S. Shen, S. Elhatisari, T.A. Lahde, D. Lee, B.-N. Lu and U.G. Meissner,

This manuscript has been revised from the first version, including some additional calculations. The conclusion in the main text is still drawn in terms of the SU(4) model with the corresponding model interaction: the treatment of the nuclear forces is simplified down to the form given by eq. (1), where the interaction comprises the quadratic and cubic terms of a density. This simplification is most likely one of the driving forces leading to the conclusion that the alpha clusters are configured in the "equilateral triangle formation" or the obtuse one with large angles mostly > 150 degrees (Fig. 2 of the manuscript). I found that the basic physics has not changed from the first version, although some arguments have been added referring to the N2LO chiral EFT interaction mainly in the supplemental part.

We can point out that some papers have already shown different conclusions based on more realistic interactions. For instance, Ref. [25] (of the manuscript) presents the following density profile in its fig. 2,

FIG. 1 of the Supplementary file attached.

First of all, the triangle configuration is not new. Secondly, none of these figures show equilateral or large-angle (> 90 degrees) obtuse triangle configurations. Ref. [25] explicitly states "Two of the alpha particles are typically close to each other and the third one is farther away". The results of Ref. [25] were obtained by the FMD method with realistic interaction, and are considered to produce more accurate results, in comparison to experiment, than the present manuscript. A similar property is depicted in Ref. [53], the very last reference added only in the revision. As the title of this reference, "alpha-Clustering in atomic nuclei from first principles with statistical learning and the Hoyle state character", indicates, this reference exhibits results of first-principles calculations with the N3LO chiral EFT interaction without approximations like the SU(4) truncation. Note that the added argument in the revised version of the present manuscript is based on the N2LO interaction, which is one-order lower than the N3LO. So, it is an understatement that Ref. [53] should be more plausible than the present manuscript, and we found that no equilateral triangular formation appears there. Ref. [53] indicates that primary basis vectors for the Hoyle state show clear triangle configurations as consequences of nuclear forces, but the opening angles are mostly < 90 degrees, as exemplified in the figure below

taken from Ref. [53]. This feature remains in the Q-aligned scheme.

FIG. 2 of the Supplementary file attached.

Thus, the main feature of the present manuscript, characterized by equilateral and large-angle obtuse triangles, is not shared by realistic calculations. The equilateral triangle can be interpreted as a direct consequence of the simple interaction depending only on the smeared density: an optimal distance between two clusters dominates the structure, for instance. The large-angle triangles are to be interpreted, and can be an intriguing subject of this toy model, even if it is an artifact. The authors claim that the GFMC approach in Ref. [22] showed the same conclusions as the present manuscript, but Ref. [22] actually mentioned them as "possible interpretation", no definite statement at all. Thus, the present manuscript exhibits the results of a simple model, and the results differ from more reliable earlier works. Other works reproduce experimental data as well. We also note, along this line, that the authors' claim "first model-independent density projection" is not founded. The authors must indicate what are differences from existing *ab initio* approaches for the intrinsic densities, for instance, VMC (GFMC) (Wiringa et al., PRC62, 014001, etc.), MCSM (Ref. [53]), ..

Many inadequate statements are found in the abstract alone. First, this work is far from "ab initio framework" because of the SU(4) truncation. If the authors want to claim any link to *ab initio* approaches, a comparison to the existing results, such as those by GFMC, NCSM and MCSM, needs to be presented in a table in the MAIN text with the descriptions of similarities and differences also in the main text.

An example of other inadequate expressions is "Some nuclear states of ^{12}C can be preferentially treated as a collection of independent particles held by the mean field of the nucleus, while other states behave more as a collection of three alpha-particle clusters". None of nuclear states are in such extreme structures, and their mixing is not excluded a priori.

It is very strange and even unfair that Ref. [53] is quoted only at the end of the manuscript. This is also inconvenient to the reader. The work shown in Ref. [53] is certainly of higher quality, and it covered the same subjects in a comprehensive way. Ref. [53] must be cited in the same way as Refs. [18-33]. The authors are aware that it was published before the first submission of the present manuscript. A similar argument may apply to Refs. [51,52].

The present work is a simple model/truncation to realistic calculations, and more advanced calculations have already been published. The present manuscript does not go beyond such earlier works, and is rather technical. Nevertheless, many expressions are misleading. Although the group of Prof. Meissner and Prof. Dean have published excellent works on related subjects, this manuscript is different. I cannot recommend the publication of this manuscript in *Nature Communications*, as it is intended to supply papers of high scientific quality to general audience. The appearance of simple structure such as equilateral triangle due to the simplification of the theory setup, as well as its fading, might attract some technical interest, as somewhat enhanced by the added part in the revision. Such a technical report might fit specialized journals.

Reviewer #1

After reading the author response and the revised version of the manuscript I can confirm that the authors have modified the manuscript accordingly to my comments and suggestions. Therefore, I can recommend the paper for publication in Nature communications.

We thank Reviewer #1 for the positive comments and recommendation for publication in Nature Communications.

Reviewer #2

I am happy that the authors have addressed questions I raised and comments that I made. I believe this work is of high importance and could be accepted for publication. The authors have worked hard to demonstrate the rigor of their approach.

We thank Reviewer #2 for the positive comments, appreciation of the high importance of the work and effort expended at demonstrating rigor, and recommendation for publication.

Reviewer #3

This manuscript has been revised from the first version, including some additional calculations. The conclusion in the main text is still drawn in terms of the $SU(4)$ model with the corresponding model interaction: the treatment of the nuclear forces is simplified down to the form given by eq. (1), where the interaction comprises the quadratic and cubic terms of a density.

To emphasize the importance of the new *ab initio* calculations performed using chiral effective field theory interactions at N²LO and to counter the statement by Reviewer #3 that our conclusions are based on the $SU(4)$ interaction, we have moved the N²LO chiral results for the structure of the ground state and the Hoyle state into the main text.

This simplification is most likely one of the driving forces leading to the conclusion that the alpha clusters are configured in the “equilateral triangle formation” or the obtuse one with large angles mostly > 150 degrees (Fig. 2 of the manuscript). I found that the basic physics has not changed from the first version, although some arguments have been added referring to the N²LO chiral EFT interaction mainly in the supplemental part.

The primary results of this work have been confirmed by the N²LO chiral results. The alpha cluster structures and geometry do emerge from the chiral N²LO calculations. The suggestion by Reviewer #3 that the simplified interactions produce artificial cluster structures is interesting. However, the chiral N²LO results show that this speculation is without merit.

We can point out that some papers have already shown different conclusions based on more realistic interactions. For instance, Ref. [25] (of the manuscript) presents the following density profile in its fig. 2. First of all, the triangle configuration is not new. Secondly, none of these figures show equilateral or large-angle (> 90 degrees) obtuse triangle configurations. Ref. [25] [Chernyk et al., Phys. Rev. Lett. 98, 032501 (2007)] explicitly states “Two of the alpha particles are typically close to each other and

the third one is farther away". The results of Ref. [25] were obtained by the FMD method with realistic interaction, and are considered to produce more accurate results, in comparison to experiment, than the present manuscript.

[redacted]

This figure shows basis-state density snapshots from Chernyk et al., Phys. Rev. Lett. 98, 032501 (2007) using the UCOM formalism (unitary correlation operator method) and FMD (fermionic molecular dynamics). It is not an *ab initio* calculation since phenomenological changes were made to improve the bound state properties of the underlying interaction, and the wave functions are Slater determinants built on single-particle wave packets of Gaussian shape. These snapshots give a rough qualitative impression of the structure of ground state and Hoyle state. Unfortunately, such plots have no rigorous scientific meaning. Almost any set of density snapshots for the important basis states can be synthesized from the same underlying wave function $|\psi\rangle$.

To demonstrate the problem, we consider a simple five-dimensional linear space defined on five lattice sites of a one-dimensional chain, which we denote as $|1\rangle, |2\rangle, |3\rangle, |4\rangle, |5\rangle$. We choose the underlying wave function $|\psi\rangle$ to be $|3\rangle$, and the corresponding density is a Kronecker delta function located at the middle lattice site, 3.

We now consider a basis set defined as $\{|1\rangle, |2\rangle, |4\rangle, 1/\sqrt{2}(|3\rangle + |5\rangle), 1/\sqrt{2}(|3\rangle - |5\rangle)\}$. The corresponding basis-state density snapshots are shown in the plots below. There are two basis states that have nonzero overlap with $|\psi\rangle$, and neither one has a density plot similar to the exact density plot for $|\psi\rangle$.

We next consider a Fourier basis defined as

$$\left\{ \begin{aligned} &1/\sqrt{5} (|1\rangle + |2\rangle + |3\rangle + |4\rangle + |5\rangle), \\ &1/\sqrt{5} (|1\rangle + e^{i\theta_5} |2\rangle + e^{i2\theta_5} |3\rangle + e^{i3\theta_5} |4\rangle + e^{i4\theta_5} |5\rangle), \\ &1/\sqrt{5} (|1\rangle + e^{i2\theta_5} |2\rangle + e^{i4\theta_5} |3\rangle + e^{i\theta_5} |4\rangle + e^{i3\theta_5} |5\rangle), \\ &1/\sqrt{5} (|1\rangle + e^{i3\theta_5} |2\rangle + e^{i\theta_5} |3\rangle + e^{i4\theta_5} |4\rangle + e^{i2\theta_5} |5\rangle), \\ &1/\sqrt{5} (|1\rangle + e^{i4\theta_5} |2\rangle + e^{i3\theta_5} |3\rangle + e^{i2\theta_5} |4\rangle + e^{i\theta_5} |5\rangle) \end{aligned} \right\},$$

where $\theta_5 = 2\pi/5$. The corresponding basis-state density snapshots produce the plots shown below. This choice of basis produces much different basis-state density snapshots. Furthermore, the basis-state density snapshots bear no relation with the exact density.

We have thus demonstrated that basis-state density snapshots have no rigorous scientific meaning. In our examples, we have chosen to use orthogonal basis states. An even more substantial distortion of the basis-state density snapshots can be made using non-orthogonal basis states. This can be seen by taking any orthogonal basis set

$$\{|b_1\rangle, |b_2\rangle, \dots |b_N\rangle\}. \quad (1)$$

We write the basis decomposition of the wave function $|\psi\rangle$ as

$$|\psi\rangle = \sum_j c_j |b_j\rangle. \quad (2)$$

We now choose some arbitrary basis state $|b_k\rangle$ and replace it by two non-orthogonal basis states, $|b_k^+\rangle = \varepsilon |b_k\rangle + |v\rangle$ and $|b_k^-\rangle = \varepsilon |b_k\rangle - |v\rangle$, for some arbitrary state $|v\rangle$ and some infinitesimal parameter ε . We note that

$$|b_k\rangle = \frac{1}{2\varepsilon} [|b_k^+\rangle + |b_k^-\rangle]. \quad (3)$$

Using the non-orthogonal basis, the new basis decomposition is

$$|\psi\rangle = \sum_{j \neq k} c_j |b_j\rangle + \frac{c_k}{2\varepsilon} [|b_k^+\rangle + |b_k^-\rangle]. \quad (4)$$

By taking the limit $\varepsilon \rightarrow 0$, the coefficients of $|b_k^+\rangle + |b_k^-\rangle$ can be made arbitrarily large. In the limit $\varepsilon \rightarrow 0$, we note that $|b_k^+\rangle \rightarrow |v\rangle$ and $|b_k^-\rangle \rightarrow |v\rangle$. We conclude that the density snapshots for the most important basis states are completely arbitrary and can be made to correspond to any arbitrary state $|v\rangle$ of our choosing.

We note that non-orthogonal basis-state snapshots were used to create Fig. 2 in Chernyk et al., Phys. Rev. Lett. 98, 032501 (2007). Non-orthogonal basis-state snapshots are also used frequently in the Monte Carlo Shell Model literature, including the recent paper, Otsuka et al., Nat. Comm. 13, 2234 (2022). Our analysis shows that such plots are lacking rigorous scientific content.

A similar property is depicted in Ref. [53] [Otsuka et al., Nat. Comm. 13, 2234 (2022)], the very last reference added only in the revision. As the title of this reference, “alpha-Clustering in atomic nuclei from first principles with statistical learning and the Hoyle state character”, indicates, this reference exhibits results of first-principles calculations with the N3LO chiral EFT interaction without approximations like the SU(4) truncation.

FIG. 2

Our discussion above has demonstrated that basis-state density snapshots are lacking rigorous scientific content, especially for the non-orthogonal basis states used in Otsuka et al., Nat. Comm. 13, 2234 (2022).

In contrast, the model-independent results presented in our work are based on the full normal-ordered A -body density distribution

$$\langle \Psi | : \rho(\mathbf{r}_1) \rho(\mathbf{r}_2) \cdots \rho(\mathbf{r}_A) : | \Psi \rangle. \quad (5)$$

These A -body densities are observables that can in principle be measured in the laboratory. We note that such A -body density distributions have not been computed using other *ab initio* methods due to computational difficulties. This points to the “high importance” (quoting Reviewer #2) of our submission to Nature Communications.

The criticism of the SU(4) interaction is misleading since our findings have also been confirmed using chiral N2LO interactions.

Note that the added argument in the revised version of the present manuscript is based on the N2LO interaction, which is one-order lower than the N3LO.

We have new N3LO calculations which further confirms our N2LO results. See Elhatisari et al., arXiv:2210.17488.

So, it is an understatement that Ref. [53] should be more plausible than the present manuscript, and we found that no equilateral triangular formation appears there. Ref. [53] indicates that primary basis vectors for the Hoyle state show clear triangle configurations as consequences of nuclear forces, but the opening angles are mostly < 90 degrees, as exemplified in the figure below taken from Ref. [53]. This feature remains in the Q -aligned scheme.

In the discussion above, we have shown that the basis-state density snapshots give arbitrary results. We now show that the Q -aligned scheme is also give arbitrary results. We pointed out the problems in our first reply to the referees. We restate the conceptual problems again here.

In Otsuka et al., Nat. Comm. 13, 2234 (2022), the intrinsic density is defined in terms of a Q -aligned state where each basis state in the many-body wave function is rotated according to their principal axes. Unfortunately, the Q -aligned state definition depends on the specific choice of many-body basis states and nearly any final result can be obtained by choosing different basis states.

The problem with the Q -aligned state definition can be seen easily from a simple model of two bosons with equal masses interacting in three dimensions via a two-body potential that is a quadratic function of the separation distance between particles. Let x, y, z be the coordinates for the relative separation vector between the two particles. The ground state of the system will have a three-dimensional wave function of the form

$$\Psi(x, y, z) = \frac{1}{\pi^{3/4} R^{3/2}} \exp \left[-\frac{x^2}{2R^2} - \frac{y^2}{2R^2} - \frac{z^2}{2R^2} \right]. \quad (6)$$

Let us now perform the basis alignment procedure using two different sets of bases. The first basis

we use is the position eigenstate basis. If we use the position space basis and align the largest principal axis along the z -axis, with no bias distinguishing the $+z$ and $-z$ directions, we then get an intrinsic density that is extremely prolate,

$$\rho_{\text{position}}(x, y, z) = \frac{1}{\pi^{1/2}R} \exp\left[-\frac{z^2}{R^2}\right] \delta(x)\delta(y). \quad (7)$$

Suppose now we instead perform the basis alignment procedure using eigenstates of our two-body 3D harmonic oscillator. In that case we need only one basis state corresponding to our ground state wave function itself. The resulting intrinsic density is then spherical,

$$\rho_{\text{harmonic}}(x, y, z) = \frac{1}{\pi^{3/2}R^3} \exp\left[-\frac{x^2}{R^2} - \frac{y^2}{R^2} - \frac{z^2}{R^2}\right]. \quad (8)$$

Clearly the Q -aligned state definition does not allow for a model-independent description of intrinsic densities and shapes.

We now present a second example showing the model dependence of the Q -aligned state definition. Consider a wave function composed of a single basis state $|b\rangle$. For simplicity, we assume that $|b\rangle$ is already Q -aligned so that no rotation is needed. We now pick an arbitrary state $|v\rangle$ such that the orientation of the principal axes for $|v\rangle$ is different from that of the state $|b\rangle - |v\rangle$. We now choose a basis set that includes the basis states $|v\rangle$ and $|b\rangle - |v\rangle$. Next we perform the Q -alignment of $|b\rangle$ using the basis decomposition $|b\rangle = |v\rangle + (|b\rangle - |v\rangle)$. Since $|v\rangle$ and $|b\rangle - |v\rangle$ have different principal axes, they must be rotated differently during the Q -alignment process. Hence, the Q -aligned state obtained using this new basis decomposition is different from $|b\rangle$. We see immediately that the Q -aligned state definition is not well defined.

*Thus, the main feature of the present manuscript, characterized by equilateral and large-angle obtuse triangles, is not shared by realistic calculations. The equilateral triangle can be interpreted as a direct consequence of the simple interaction depending only on the smeared density: an optimal distance between two clusters dominates the structure, for instance. The large-angle triangles are to be interpreted, and can be an intriguing subject of this toy model, even if it is an artifact. The authors claim that the GFMC approach in Ref. [22] showed the same conclusions as the present manuscript, but Ref. [22] actually mentioned them as “possible interpretation”, no definite statement at all. Thus, the present manuscript exhibits the results of a simple model, and the results differ from more reliable earlier works. Other works reproduce experimental data as well. We also note, along this line, that the authors’ claim “first model-independent density projection” is not founded. The authors must indicate what are differences from existing *ab initio* approaches for the intrinsic densities, for instance, VMC (GFMC) (Wiringa et al., PRC62, 014001, etc.), MCSM (Ref. [53]), ..*

In addition to the Green’s function Monte Carlo calculations of Carlson et al., Rev. Mod. Phys. 87 1067 (2015), these three independent *ab initio* calculations are in excellent agreement with the findings of our Nature Communications submission: Epelbaum et al., Phys. Rev. Lett 106, 192501 (2011); Epelbaum et al., Phys. Rev. Lett. 109, 252501 (2012); and Dreyfuss et al., Phys. Lett. B 727, 511 (2013).

Reviewer #3 suggests comparing our $A = 12$ carbon results with that of Wiringa et al., Phys. Rev. C 62, 014001 (2000), which is entitled “Quantum Monte Carlo calculations of $A = 8$ nuclei”. This is not a viable suggestion since the nuclei are different. Reviewer #3 also suggests comparing with Monte Carlo

Shell Model results. As discussed above, this is not a viable suggestion due to the arbitrary nature of the basis-state density snapshots and the Q -aligned state formalism.

Given the model-dependent nature of basis-state density snapshots, the Q -aligned state formalism, and other results presented in previous works, we believe that the description “first model-independent density projection” is accurate.

Many inadequate statements are found in the abstract alone. First, this work is far from “ab initio framework” because of the $SU(4)$ truncation.

As discussed above, this statement is incorrect.

If the authors want to claim any link to ab initio approaches, a comparison to the existing results, such as those by GFMC, NCSM and MCSM, needs to be presented in a table in the MAIN text with the descriptions of similarities and differences also in the main text.

To emphasize the importance of the new *ab initio* calculations performed using chiral effective field theory interactions at N2LO, we have moved the N2LO chiral results for the structure of the ground state and the Hoyle state into the main text.

The comparison with MCSM intrinsic shape results from Otsuka et al., Nat. Comm. 13, 2234 (2022), is not possible due to the arbitrary nature of those results. See the discussion above regarding the VMC results of Wiringa et al., Phys. Rev. C 62, 014001 (2000). We have already included a discussion of the agreement with GFMC results from Carlson et al., Rev. Mod. Phys. 87 1067 (2015).

An example of other inadequate expressions is “Some nuclear states of ^{12}C can be preferentially treated as a collection of independent particles held by the mean field of the nucleus, while other states behave more as a collection of three alpha-particle clusters”. None of nuclear states are in such extreme structures, and their mixing is not excluded a priori.

To address the concerns of Reviewer #3, we have now written “Some nuclear states of ^{12}C can be preferentially treated as mostly a collection of independent particles held by the mean field of the nucleus, while other states behave more as a collection of three alpha-particle clusters. But these two pictures are not mutually exclusive. Some mixing is possible and some states can be described in either fashion.”

It is very strange and even unfair that Ref. [53] is quoted only at the end of the manuscript. This is also inconvenient to the reader. The work shown in Ref. [53] is certainly of higher quality, and it covered the same subjects in a comprehensive way. Ref. [53] must be cited in the same way as Refs. [18-33]. The authors are aware that it was published before the first submission of the present manuscript. A similar argument may apply to Refs. [51,52].

As pointed out above, the quality of the Otsuka et al., Nat. Comm. 13, 2234 (2022), work for describing the intrinsic shapes of the ground state and Hoyle state is disputable. But to address the concerns of Reviewer #3, we have now cited Otsuka et al. twice, with the first instance appearing early in the manuscript.

As of this writing, Otsuka et al., Nat. Comm. 13, 2234 (2022), was never posted on arXiv. It was published on April 27, 2022. Our original submission was to Nature on March 8, 2022. It was eventually transferred to Nature Communications on March 31, 2022. Reviewer #3 is factually incorrect.

We agree that the other citations could have appeared in the original manuscript. This was corrected in the first revision.

The present work is a simple model/truncation to realistic calculations, and more advanced calculations have already been published. The present manuscript does not go beyond such earlier works, and is rather technical. Nevertheless, many expressions are misleading. Although the group of Prof. Meissner and Prof. Dean have published excellent works on related subjects, this manuscript is different. I cannot recommend the publication of this manuscript in Nature Communications, as it is intended to supply papers of high scientific quality to general audience. The appearance of simple structure such as equilateral triangle due to the simplification of the theory setup, as well as its fading, might attract some technical interest, as somewhat enhanced by the added part in the revision. Such a technical report might fit specialized journals.

The first sentence is incorrect, and we disagree with this assessment. We concur with Reviewer #1 and Reviewer #2 that the manuscript is ready for publication in Nature Communications.

REVIEWERS' COMMENTS

Reviewer #3 (Remarks to the Author):

The referee noticed that the manuscript has been revised particularly in highlighting the fact that the interactions are SU(4) and N2LO. As the usual ab initio calculations take higher-order interactions (N3LO, ...), this clear remark is important to distinguish this work from them. Although the authors are reluctant to accept, this referee believes that simple structures in the presently obtained theoretical results are largely due to this simplification, not due to the nature. This referee noticed the difference between SU(4) and N2LO density profiles, having known that the reality is beyond the N2LO. The NN scattering phase shifts cannot be reproduced well at the level of N2LO, and nuclear properties need correct phase shifts in order to be described well.

The citations have been corrected properly, although this could have been done in the first revision.

Most of the arguments in the reply regarding the Q-alignment are irrelevant, and are not precise enough. Particularly, before the summary section of the manuscript, there remains a sentence, "... not possible because the Q-aligned state definition depends on the specific choice of many-body basis states." This sentence is found to be very misleading. Generally and presently, the CI calculations are model independent, and the wavefunctions were tested from different angles in the quoted reference. In order to be precise, the sentence should be changed, for instance, as "... not possible at present." The referee also noticed that the quantum fluctuations of the intrinsic density profile are discussed in the quoted reference. So it is not a simply matter of the so-called Q-alignment.

FMD density profiles are not non-scientific, contrary to the authors' reply. But this claim is not mentioned in the manuscript.

This is all this referee can report.

Reviewer #3 (Remarks to the Author):

*The referee noticed that the manuscript has been revised particularly in highlighting the fact that the interactions are $SU(4)$ and N2LO. As the usual *ab initio* calculations take higher-order interactions (N3LO, ...), this clear remark is important to distinguish this work from them. Although the authors are reluctant to accept, this referee believes that simple structures in the presently obtained theoretical results are largely due to this simplification, not due to the nature. This referee noticed the difference between $SU(4)$ and N2LO density profiles, having known that the reality is beyond the N2LO. The NN scattering phase shifts cannot be reproduced well at the level of N2LO, and nuclear properties need correct phase shifts in order to be described well.*

The citations have been corrected properly, although this could have been done in the first revision.

Most of the arguments in the reply regarding the Q-alignment are irrelevant, and are not precise enough. Particularly, before the summary section of the manuscript, there remains a sentence, "... not possible because the Q-aligned state definition depends on the specific choice of many-body basis states." This sentence is found to be very misleading. Generally and presently, the CI calculations are model independent, and the wavefunctions were tested from different angles in the quoted reference. In order to be precise, the sentence should be changed, for instance, as "... not possible at present." The referee also noticed that the quantum fluctuations of the intrinsic density profile are discussed in the quoted reference. So it is not a simply matter of the so-called Q-alignment.

FMD density profiles are not non-scientific, contrary to the authors' reply. But this claim is not mentioned in the manuscript.

This is all this referee can report.

We thank Reviewer #3 for this suggestion. The requested change has been made to the text.